# Effects of the Weight and Balance of Head-Mounted Displays on Physical Load

**Kodai Ito** [1,]*[ ], **Mitsunori Tada** [2][ ], **Hiroyasu Ujike** [2][ ] **and Keiichiro Hyodo** [2]

[1] College of Science and Engineering, Aoyama Gakuin University, Kanagawa 252-5258, Japan
[2] National Institute of Advanced Industrial Science and Technology, Tokyo 135-0064, Japan;
m.tada@aist.go.jp (M.T.); h.ujike@aist.go.jp (H.U.); kei.hyodo@yuasa-system.jp (K.H.)
* Correspondence: kodai.ito@it.aoyama.ac.jp

**Abstract:** To maximize user experience in VR environments, optimizing the comfortability of head-mounted displays (HMDs) is essential. To date, few studies have investigated the fatigue induced by wearing commercially available HMDs. Here, we focus on the effects of HMD weight and balance on the physical load experienced by the user. We conducted an experiment in which participants completed a shooting game while wearing differently weighted and balanced HMDs. Afterwards, the participants completed questionnaires to assess levels of discomfort and fatigue. The results clarify that the weight of the HMD affects user fatigue, with the degree of fatigue varying depending on the center of mass position. Additionally, they suggest that the torque at the neck joint corresponds to the physical load imparted by the HMD. Therefore, our results provide valuable insights, demonstrating that, to improve HMD comfortability, it is necessary to consider both the balance and reduction of weight during HMD design.

**Keywords:** head mounted display; physical load; product design

## 1. Introduction

For positive experiences using virtual reality (VR) systems, user comfort when wearing head-mounted displays (HMDs) is essential. Recently, rapid advances in VR technology has led to the increased availability of VR experiences [1,2]. In addition to significant commercial interest from the entertainment industry, VR technology is used as an image display device in various fields such as design and engineering. However, few studies have investigated the fatigue induced by wearing commercially available HMDs. The weight and balance of HMDs are key factors impacting user comfort and affecting the head movements and physical loading experienced by HMD users. Therefore, these factors must be considered when seeking to improve the comfortability of HMDs.

The authors in [3,4] suggested that increasing the mass of an HMD and the position of its center of mass (COM) may affect the physical workload. There is a relationship between the moment and the workload on the neck and lumbar joints. In the field of aviation, the authors in [5–7] reported the relationship between the physical features of helmet-mounted displays and the workload on the neck. In particular, the authors in [7] discussed the maximum torque that can be tolerated by the neck. For long-term use, it is necessary to reduce the weight of the HMD or optimize the COM position. Therefore, it is worth exploring whether similar trends apply to HMDs.

The authors in [8] evaluated the biomedical stress in the neck and shoulders due to wearing an augmented reality (AR) HMD, including electromyography (EMG) measurements of the flexion angle, moment, and muscle activity, while also considering the user-reported discomfort. Their results clarified the differences in biomechanical stresses and usability for different target distances and sizes in AR systems. Similarly, the authors in [9] focused on vertical target location with VR HMDs. They concluded that targets in the VR interface should be displayed between eye height and 15° below eye height.

Elsewhere, Chihara and Seo [10] investigated the effects of HMD weight and balance on the physical workload using a professional grade HMD (weighing 800 g with a head-worn attachment made of plastic resin). They established ranges of the mass and COM for four postures. Many lighter consumer grade HMDs are becoming available, such as the Oculus Rift CV1 (468 g) [11] and various smartphone-based HMDs [12]. These lightweight HMDs have elastic belts as the head attachment. Therefore, they should also be investigated as their weight and balance might result in more comfortable HMDs. The authors in [13] studied three different types of HMD and increased their weight to evaluate user discomfort, reporting differences between soft-belt headsets and integrated headsets. However, they only applied additional weight to the front side of the HMD. Alternatively, the authors in [14] focused on the product structures and attachment modes of AR glasses. Their results suggested that comfort varies for different attachment modes for identically weighted devices.

Some studies focused on the comfort of HMDs, but few studies comprehensively investigated weight and balance. The consumer HMDs are already made lighter in weight, but different weight balances should be considered. Chihara and Seo investigated the relationship among physical workload, weight, and balance [10]. However, the participants held the instructed postures for five seconds with professional heavy HMD. If we assume a natural HMD usage situation, the participants should get the spontaneous postures for at least a few minutes. Therefore, we decided to vary the weight and balance of the lightweight HMD to investigate the physical workload by the participant's spontaneous posture change during game play.

This study aims to clarify the relationship between the physical loads, weight, and balance of HMDs. In contrast to previous studies, we conducted experiments in which natural user situations were reproduced, such as game play, for a relatively lightweight commercially available HMD. As such, participants were instructed to play a shooting game while wearing HMDs with different weights and balances. In this game, participants shot targets that appeared at random positions and postures within their field of view. The aim of the gun moves in response to changes of the participant's posture. It took around two minutes. Following the experiment, the participants completed a questionnaire to assess the physical loading that they perceived and their comfort level. It depends on the position of the gravity force acting on the camera. In such gameplay, physical workload may depend on the position of the gravity force acting on the camera, rather than on its center of mass. In this study, we use the Center of Mass, which was used in the study by Chihara and Seo [10]. These meanings are almost the same.

## 2. Materials and Methods

### 2.1. Devices

The experiments in this study were conducted using an Oculus Rift CV1 [11], which is one of the most popular HMDs on the market. We attached hook-and-loop fasteners to the front and rear of the HMD (Figure 1), which was connected to a laptop PC (DELL: ALIEN WARE 13). The software ran at almost 90 fps.

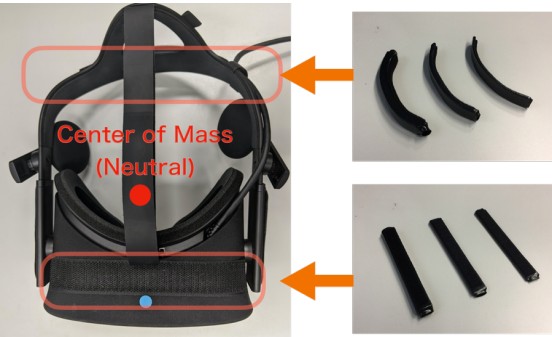

**Figure 1.** Diagram of the Oculus Rift CV1 with hook-and-loop fasteners and lead weights added.

### 2.2. Shooting Game

A shooting game was developed especially for the experiments. In this game, participants were instructed to shoot 20 targets that appeared at random positions and postures within their field of view. The aim of the gun moves in response to changes in the roll, pitch, and yaw angles of the head. To shoot a target, players had to maintain their aim for 1 s.

### 2.3. Experimental Conditions

To change the weight and balance of the HMD, we prepared lead weights that could be attached to the front and rear of the HMD (Figure 1) with the hook-and-loop fasteners. We prepared seven conditions combining weight and balance, as listed in Table 1.

The terms "front," "both," and "rear" in Table 1 indicate the lead weight attachment location. The subsequent numerical values indicate the weight of the lead weight. For example, "both 100" indicates that the HMD was loaded with 100 g lead weights both at the front and rear sides. The term "neutral" indicates that no additional weight was attached to the HMD. The values on the horizontal axis indicate the longitudinal distance of the COM from the tragion notch.

**Table 1.** Weight and balance conditions.

| Weight (g) | Longitudinal Distance Separating the Center of Mass from the Tragion Notch (mm) | | | | | | |
| | 10 | 30 | 45 | 50 | 60 | 75 | 85 |
| --- | --- | --- | --- | --- | --- | --- | --- |
| 670 | rear 200 | | both 100 | | | | front 200 |
| 570 | | rear 100 | | both 50 | | front 100 | |
| 470 | | | | | neutral | | |

The participants were separated into three groups according to the three lead weight positions, i.e., "front," "both," and "rear". For each group, one of the three positions was assigned, and, for each position, three weight conditions (i.e., neutral, 100 g, 200 g) were assigned at random, as summarized in Table 2.

**Table 2.** Groupings for the lead weight positions.

| Group | Lead Weight Position | | |
| --- | --- | --- | --- |
| front | neutral | front 100 | front 200 |
| both | neutral | both 50 | both 100 |
| rear | neutral | rear 100 | rear 200 |

### 2.4. Participants

The experiment involved 188 participants (96 males and 92 females) with ages ranging from 13 to 85 years old. Most of participants was visitors to the exhibition and voluntarily participated. The shortage of participants was recruited through a recruiting company for a fee. The experiment was conducted with the approval of the Ethical Review Committee, and the data were anonymized for statistical analysis. Table 3 lists the number of participants grouped by gender, age, and weight positions.

**Table 3.** Details of participants.

| Gender | Group | 10 s | 20 s | 30 s | 40 s | 50 s | 60 s | 70 s | 80 s | Total |
|--------|-------|------|------|------|------|------|------|------|------|-------|
| Male | Front | 5 | 5 | 4 | 8 | 5 | 4 | 4 | 1 | 36 |
| | Both | 3 | 5 | 2 | 7 | 5 | 4 | 4 | 0 | 30 |
| | Rear | 3 | 6 | 1 | 7 | 5 | 4 | 4 | 0 | 30 |
| **Male total** | | 11 | 16 | 7 | 22 | 15 | 12 | 12 | 1 | 96 |
| Female | Front | 3 | 8 | 3 | 7 | 5 | 3 | 5 | 0 | 34 |
| | Both | 3 | 6 | 2 | 5 | 4 | 4 | 4 | 0 | 28 |
| | Rear | 3 | 7 | 3 | 6 | 3 | 4 | 4 | 0 | 30 |
| **Female total** | | 9 | 21 | 8 | 18 | 12 | 11 | 13 | 0 | 92 |
| **Total** | | 20 | 37 | 15 | 40 | 27 | 23 | 25 | 1 | 188 |

## 3. Measurements

### 3.1. Physical Properties

We measured the pupillary distance and grip strength of the participants. In addition, we asked the participants to self-assess their visual acuity. Pupillary distance and visual acuity were measured to verify low visual acuity or pupillary distance adjustment would interfere with game play. Grip strength was measured as a surrogate for the participant's total body muscle mass.

### 3.2. Data Collected by the HMD

The HMD recorded the position and posture of the targets in the shooting game and the corresponding head movements of the participants (i.e., the roll, pitch, and yaw angles). In addition, the time taken for each participant to complete the game was recorded. These were measured to verify that the game was not progressing due to abnormal behavior.

### 3.3. Questionnaires

#### 3.3.1. Physical Load

The participants were instructed to complete this questionnaire immediately after game completion for each condition. They were asked to evaluate the HMD fixation, video clarity, video fatigue, weight fatigue, and balance fatigue using five-point Likert scales. Our questionnaire is shown in the Appendix A. The first two questions were asked to verify whether the presentation of the game images was appropriate. The latter three questions were asked to assess subjective physical workload.

#### 3.3.2. Residual Effect

The participants were instructed to complete this questionnaire 15 min after finishing the last game. In this case, they were asked to evaluate residual eye discomfort, increased eye discomfort, residual head discomfort, increased head discomfort, residual neck discomfort, and increased neck discomfort using five-point Likert scales. These questions were asked to verify that the physical workload caused by the weight used in this study would not cause sequelae.

## 4. Results

Based on the measurement results of visual acuity and pupillary distance, all subjects could see our game image normally. In addition, we verified the data collected by the HMD, and there was no abnormal behavior in the game.

The results of grip strength measurement showed that male participants had a stronger grip strength than female participants on average, with male participants ranging from 40–50 years old exhibiting the strongest grip strength. We separated the participants into two groups based on their average grip strength and compared the results of the questionnaires between these groups. However, no significant differences were observed

from this comparison. Next, we separated the participants into two mixed-gender groups according to their age: young (13 to 49 years old) and elderly (50 to 85 years old).

In the following sections, we performed statistical analysis. ANOVA was performed with data that have a normal distribution and equal variances.

*4.1. Game Completion Times*

Figure 2 shows the game completion time for male participants, while Figure 3 shows the corresponding results for female participants. Each figure presents the results for the "front," "both," and "rear" groups. In addition, we separated the results into the young (13–49 years old) and the elderly (50–85 years old) groups.

For each scale, we conducted a three-factor mixed ANOVA on age, gender, and weight condition and identified significant age-related trends in each case. The elderly group took longer to complete the game than the young group for most conditions (front: $F(1,198) = 34.366$, $p = 0.000$, both:$F(1,168)$ 7.845, $p = 0.006$, rear:$F(1,174) = 48.658$, $p = 0.000$). In contrast, our analyses of the weight condition and gender factors did not reveal notable trends.

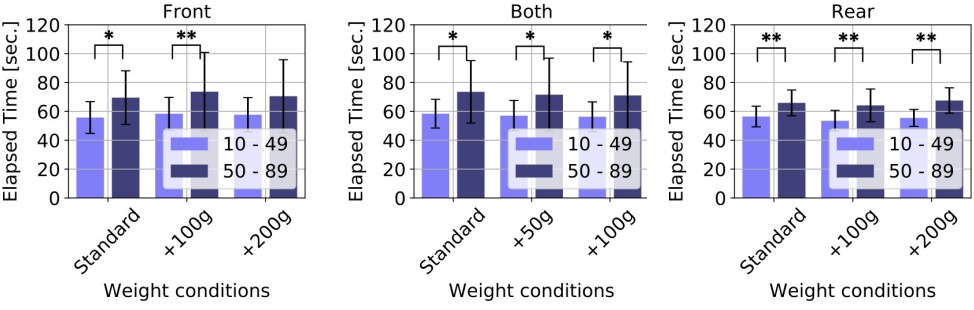

**Figure 2.** Game completion time for male participants. (*: $p < 0.05$, **: $p < 0.01$).

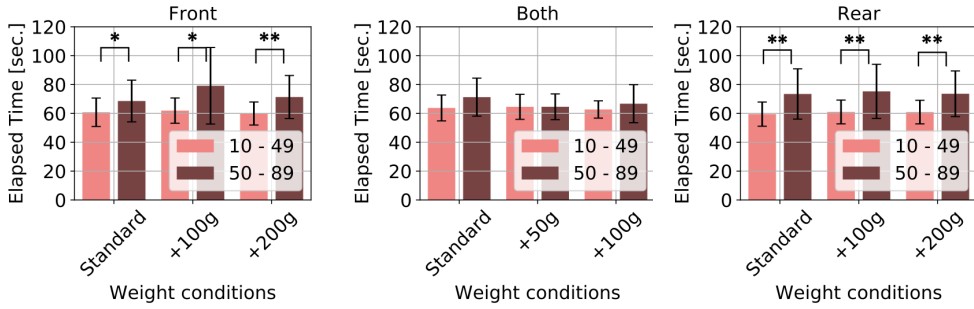

**Figure 3.** Game completion time for female participants. (*: $p < 0.05$, **: $p < 0.01$).

*4.2. Questionnaire Results*

4.2.1. Physical Load

The questionnaire results providing subjective feedback regarding the physical load are shown for the "front" (Figures 4 and 5), "both" (Figures 6 and 7), and "rear" (Figures 8 and 9) groups.

On average, the participants gave HMD fixation to the head about a score of 4 (i.e., slightly agree), indicating that most participants did not feel discomfort. We conducted a three-factor mixed ANOVA for each scale on age, gender, and weight condition and identified significant weight condition-related trends for "front" group ($F(2,198) = 5.402$, $p = 0.005$). Age only emerged as a factor in the analysis of the "Both" group ($F(1,168) = 6.491$, $p = 0.012$). The participants rated the sharpness of the graphics approximately 4 on average (slightly agree), indicating that most participants did not suffer significant focus-related issues during game play. We conducted the same ANOVA, which identified the significance

of the weight condition factor for the "front" group (F(2,198) = 5.851, *p* = 0.005) and the age factor for the "both" group (F(1,168) = 9.202, *p* = 0.029).

Visual fatigue was reported by some participants. The ANOVA revealed gender-related differences for the "both" and "rear" groups (front: F(1,198) = 5.402, *p* = 0.005, rear: F(1,174) = 5.402, *p* = 0.005).

Increasing the weight also induced fatigue in the participants. In this case, the ANOVA identified significant weight condition-related trends in the "front" and "both" groups (front: F(2,198) = 27.960, *p* = 0.000, both: F(2,168) = 6.849, *p* = 0.001), with these groups reporting considerable increases in fatigue as the weight added to the HMD was increased. Gender was highlighted as a factor exclusively in the "front" group (F(1,198) = 12.834, *p* = 0.000). In addition, age-related trends were observed in the "both" and "rear" groups (both: F(1,168) = 3.935, *p* = 0.049, rear: F(1,174) = 8.266, *p* = 0.005). Participants also reported feeling balance-related fatigue for some of the heavier weight conditions. The ANOVA results indicated the weight condition as a notable factor in the "front" and "both" groups (front: F(2,198) = 11.221, *p* = 0.000, both: F(2,168) = 5.402, *p* = 0.005). For these two groups, as the weight increased, feelings of fatigue increased substantially. Moreover, gender and age were revealed to be significant factors in the "front" group (gender: F(1,198) = 9.066, *p* = 0.003, age: F(1,198) = 4.757, *p* = 0.030). Age was also identified as an important factor in the "rear" group (F(1,174) = 5.622, *p* = 0.019). For the "both" group, the fatigue gradients corresponding to weight and balance were less than those for the "front" group. In contrast, the results for the "rear" group showed no significant differences between the various condition. In particular, the plots based on the responses by the young female group exhibited the possibility to be opposite gradients to those for the "front" group.

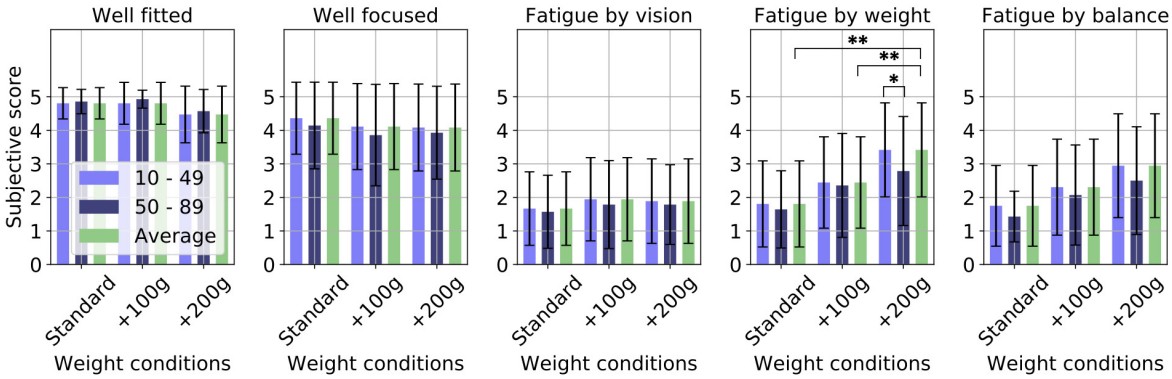

**Figure 4.** Questionnaire results for male participants in the "front" group. The results of Q1 to Q5 are shown from left to right. (\*: *p* < 0.05, \*\*: *p* < 0.01).

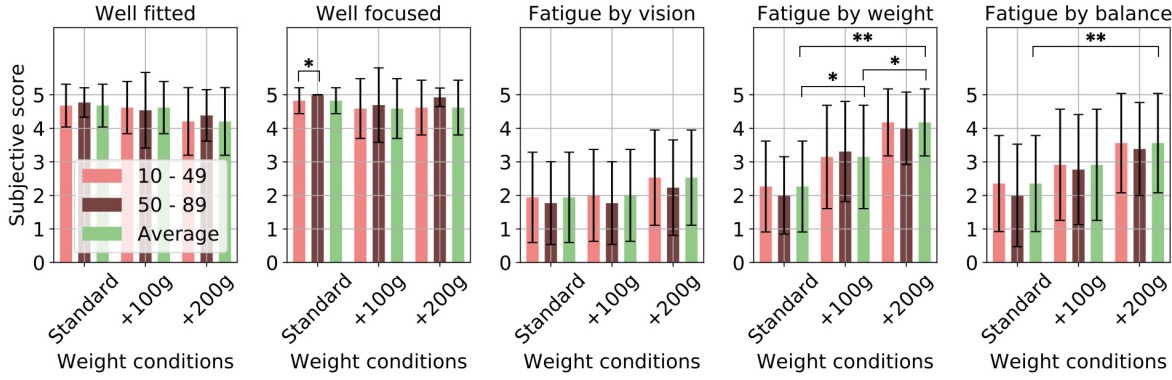

**Figure 5.** Questionnaire results for female participants in the "front" group. The results of Q1 to Q5 are shown from left to right. (\*: *p* < 0.05, \*\*: *p* < 0.01).

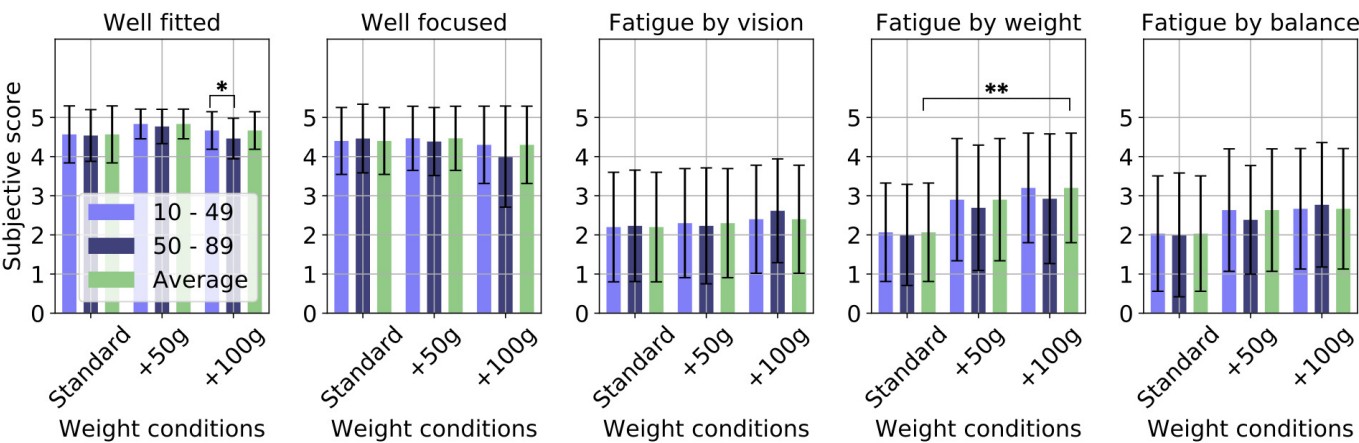

**Figure 6.** Questionnaire results for male participants in the "both" group. The results of Q1 to Q5 are shown from left to right. (*: $p < 0.05$, **: $p < 0.01$).

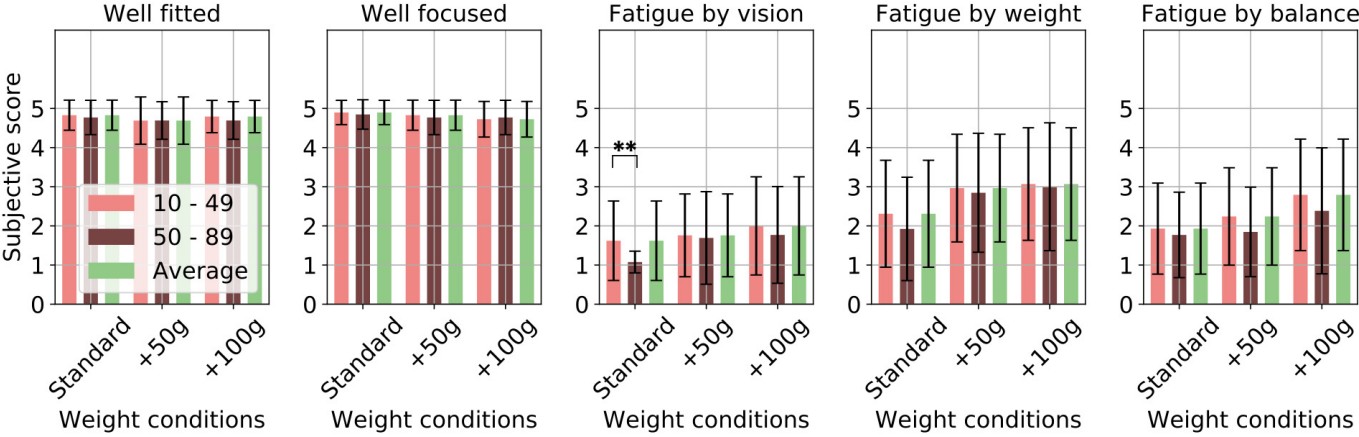

**Figure 7.** Questionnaire results for female participants in the "both" group. The results of Q1 to Q5 are shown from left to right. (**: $p < 0.01$).

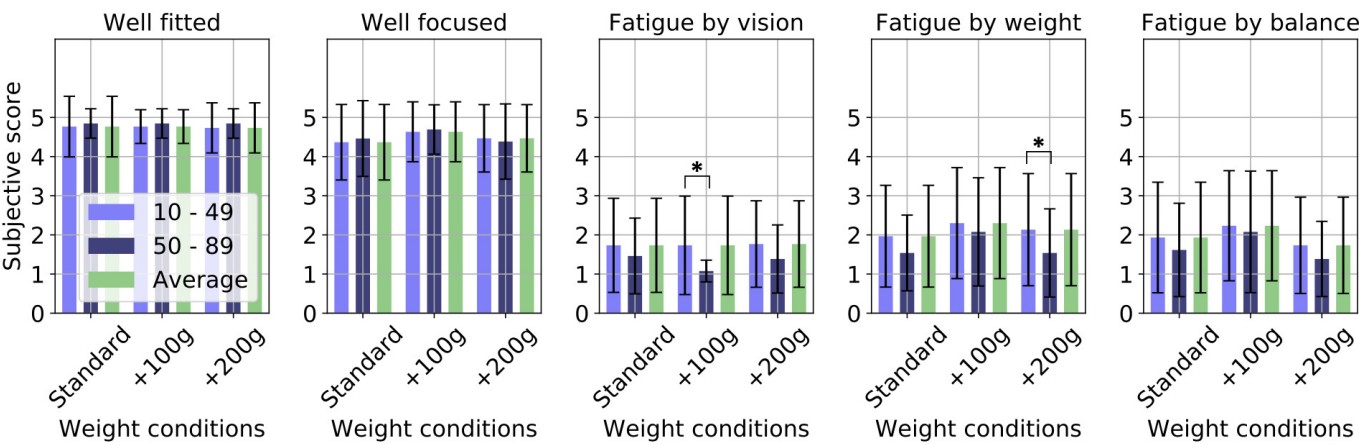

**Figure 8.** Questionnaire results for male participants in the "rear" group. The results of Q1 to Q5 are shown from left to right. (*: $p < 0.05$).

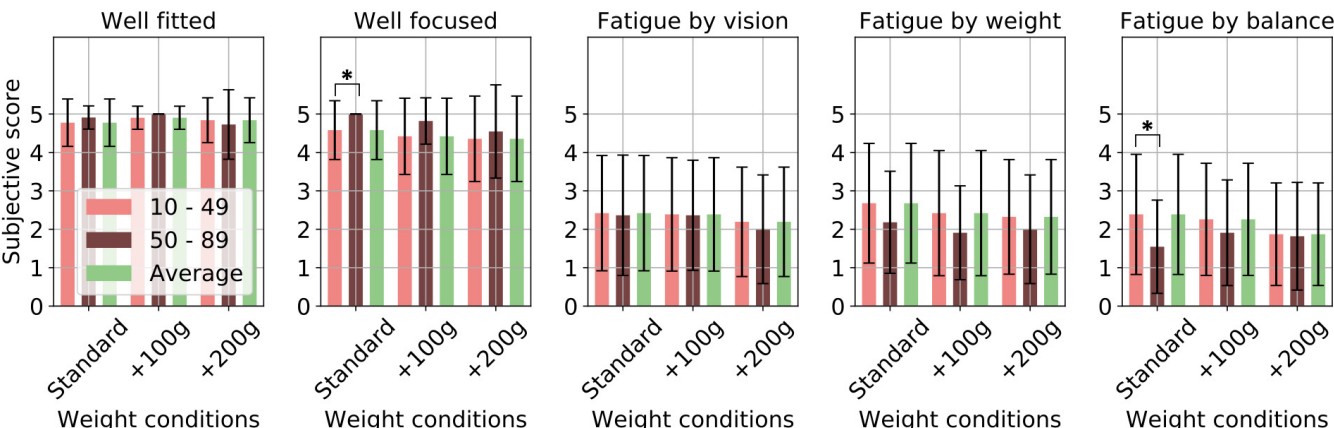

**Figure 9.** Questionnaire results for female participants in the "rear" group. The results of Q1 to Q5 are shown from left to right. (*: $p < 0.05$).

### 4.2.2. Residual Effect

Figures 10 and 11 show the results of the questionnaire answered 15 min after game completion for all three weight-location groups. On average, the participants scored the residual effects felt by the eyes, head, and neck as 2 (i.e., slightly disagree), thereby indicating insignificant residual effects.

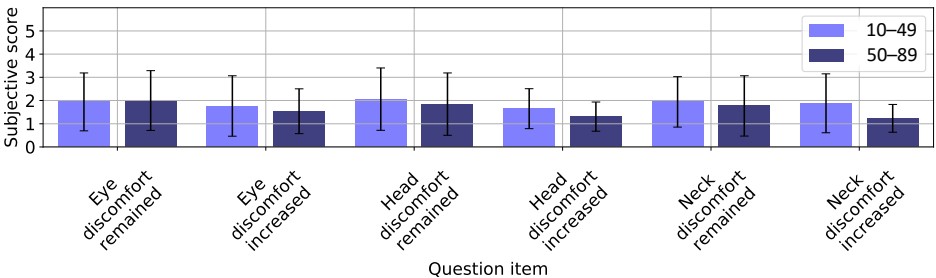

**Figure 10.** Questionnaire results 15 min after game for male participants. The results of Q1 to Q6 are shown from **left** to **right**.

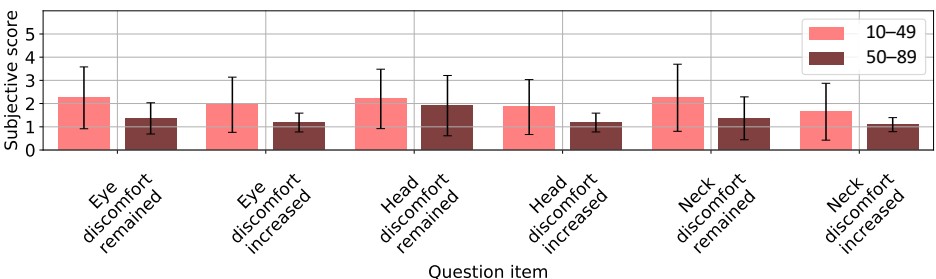

**Figure 11.** Questionnaire results 15 min after game completion for female participants. The results of Q1 to Q6 are shown from **left** to **right**.

### 4.2.3. Center of Mass

Figure 12 shows the mean fatigue score as an isosurface in two-dimensional space comprised by the weight and balance axes. The value on the vertical axis is larger as the weight increases, and on the horizontal axis is larger as the center of mass moves forward from the tragion notch. Several points determined from the experimental conditions are plotted, and the larger the fatigue score, the redder the color. These points were then linearly interpolated to form a contour plot. The blank areas were not included in the

experimental conditions. The weight and the COM conditions are listed in Table 1. The results show that, even when the added weight was unchanged, the fatigue score increased as the COM shifted toward the front of the head. In addition, even if the COM was unchanged, the fatigue score increased with the weight.

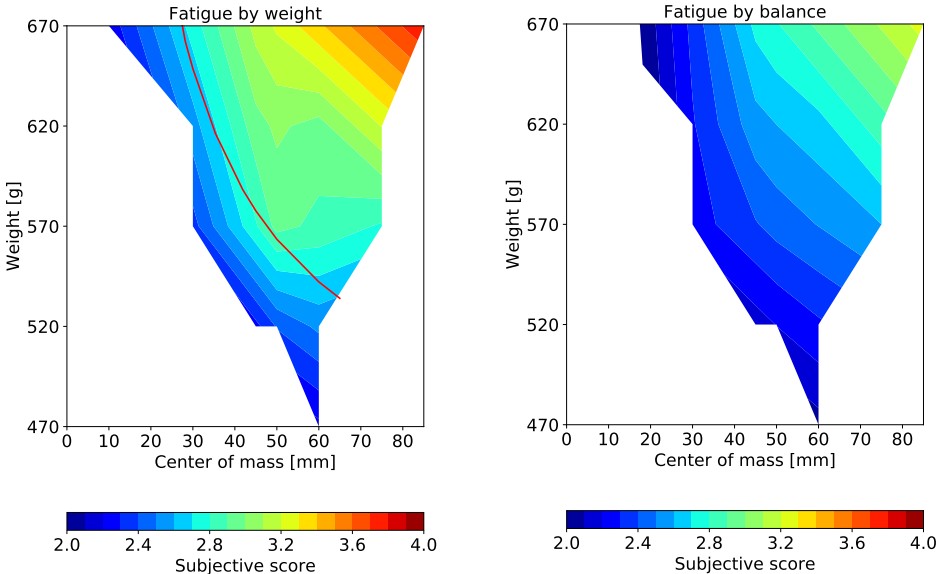

**Figure 12.** Variation in fatigue due to weight and balance.

## 5. Discussion

### 5.1. Grip Strength

As mentioned earlier, the average grip strength of male participants exceeded that of females, with the strongest grips recorded for males between 40 and 50 years old. Although neck muscle mass may be considered an important factor in the fatigue reported by participants, it could not be measured directly. Therefore, we used grip strength as a proxy. Following this approach, we identified significant gender-related differences.

### 5.2. Game Completion Time

Game completion time did not vary significantly depending on the weighting of the HMD. On average, participants took approximately two minutes to complete the game. Owing to the short game duration, weight-related effects were more subtle than they may have been for a longer task. Nevertheless, the questionnaire results relating to physical load did identify differences among the weight conditions. Therefore, for long-term use, the game completion time data may show a broader distribution. This should be clarified in future studies.

We also note that the elderly group took longer, on average, to complete the game than the young group. This is attributed to the reduced familiarity of the elderly group with playing video games. Therefore, the familiarity factor may introduce a bias that does not reflect the full extent of the effects of HMD weight and balance.

We designed this experiment to clarify the differences of the appropriate weight and balance of HMDs depending on each age. There is a bias as a consequence of their less familiarity with video games. However, if the elderly users are generally not familiar with HMD-based games, such participants should be targeted. Therefore, we considered that the bias due to lack of familiarity with games and an age bias may be inseparable.

### 5.3. Questionnaire Results

#### 5.3.1. Physical Load

The HMD fixation to the head and the sharpness of the graphics both received average scores of approximately 4 (slightly agree). Even though some significant differences

were observed between certain conditions, most of the participants did not report feeling discomfort. However, in the "front" group, the HMD fixation and graphics sharpness scores decreased as the weight was increased. It is possible that reports of imbalance are the result of ill-fitting HMDs. Importantly, the HMD fixation and graphics sharpness of HMD did not negatively impact the following results during the short task performed in this study.

However, some participants reported visual fatigue, and these reports were reasonably consistent across the various weight conditions and locations investigated in the experiment. The results show that, at least for short-term use, weight and balance do not affect visual fatigue. Reports of fatigue were more common among the elderly participants, which likely reflects their lack of familiarity with playing VR games. Similarly, elderly participants took longer to complete the game.

Questionnaires Q4 and Q5 highlighted the tendency for fatigue to increase as weight was added. This observation is supported by many previous studies on head- and helmet-mounted displays [3,5–8,10,13–15]. Evidently, reducing the weight of HMDs is beneficial for all situations.

Nevertheless, our results indicate that fatigue can be reduced by including additional weight at the rear of the HMD where possible. Because the original COM coincides with the front of the HMD, adding weight to the rear of the HMD causes the COM to shift to the center of the head, thereby reducing the physical load. Interestingly, this reduced fatigue due to rear weighting was more common among female participants. One possible explanation is that female participants had less neck muscle mass on average than their male counterparts. This is supported by the results of the grip strength test, which we used as an indicator of neck muscle mass.

### 5.3.2. Residual Effect

Few participants reported feeling residual effects after playing the game. Although participants reported feeling fatigue immediately after the task for some conditions, residual effects were minimal owing to the short game completion time. These results suggest that any fatigue associated with short-term use dissipates quickly. Further exploration is required to determine whether our findings can be extended to long-term use.

### 5.3.3. Center of Mass

Figure 12 shows a line profile resembling an inverse proportional hyperbolic curve. The Q4 curve showing a score of 2.8 can be approximated by the following equation, which shows an inversely proportional trend:

$$f(x) = 6384.3x^{-1} + 435.8 (R^2 = 0.91). \tag{1}$$

Our results suggest that fatigue will be constant if the product of the weight and the balance (i.e., the distance from the COM) is constant. The product of the weight and the balance represents the torque required to support the HMD. Thus, the torque experienced by the neck joint can be regarded as an indicator of the physical load exerted by the HMD. Guidelines for the production of helmet-mounted displays [15] show a similar relationship between weight, balance, and fatigue. Therefore, helmet- and head-mounted displays appear to exhibit certain shared characteristics.

Shifting the COM of the HMD from 2 cm to 4 cm in front of the tragion notch results in the physical load increasing rapidly. Regardless of the weight of the HMD, if the COM lies within 2 cm in front of the tragion notch, the physical load remains almost constant. Chihara and Seo [10] reported similar results for participants in a look-down posture condition. As we did not instruct participants to adopt a particular posture, they exhibited a variety in their postures. Despite these variations in user posture during our experiment, it is encouraging that professional- and consumer-use HMDs with different shapes and weights yielded similar results. By investigating content that requires upward looking

and other postures, the results reported by Chihara and Seo [10] may be applicable to consumer-use HMDs.

Our results are based on subjective responses. More objective evaluations are needed to verify the anatomical and physiological effects. For example, future work is to estimate the load torque applied to the neck joint from the posture and weights of human and HMD, and compare it with the present results.

## 6. Conclusions

In this study, we focused on the effects of HMD weight and balance on the physical load experienced by the user. We performed an experiment in which participants completed a shooting game wearing HMDs with various weights and balances. After playing the game, the participants were instructed to answer a questionnaire to assess the level of physical loading that they perceived. Notably, we found that the weight of the HMD affects perceived fatigue, with the degree of fatigue varying according to the COM. The product of the weight and the balance is equivalent to the torque required to support the HMD. Our results suggest that torque at the neck joint can be used as an indicator of the physical load imparted by the HMD. However, the limit of the torque that can be tolerated without discomfort may vary depending on the neck muscle mass of the user. Our findings indicate that situating additional weight to the rear of the HMD may benefit users with low muscle mass. In addition, user familiarity with playing games may also affect visual fatigue and task completion time. However, for short-term use, the HMD weight and balance are not significant factors affecting fatigue. In short, improving the comfortability of HMDs requires consideration of weight balance as well as weight reduction. Our results provide a valuable resource for informing future HMD design. Nevertheless, the effects of long-term use and different HMD attachments need to be studied to establish standard guidelines for comfortable HMD use.

**Author Contributions:** Conceptualization, M.T., H.U. and K.H.; Data curation, K.I. and M.T.; Formal analysis, K.I. and M.T.; Funding acquisition, H.U.; Investigation, K.I., M.T., H.U. and K.H.; Methodology, K.I. and M.T.; Project administration, M.T.; Resources, M.T., H.U. and K.H.; Software, K.I. and M.T.; Supervision, H.U.; Validation, H.U. and K.H.; Visualization, K.I.; Writing—original draft, K.I.; Writing—review and editing, M.T. All authors have read and agreed to the published version of the manuscript.

**Funding:** This study was supported by the Budget for Promotion of Strategic International Standardization, promoted by the Ministry of Economy, Trade and Industry (METI) in Japan, via Mitsubishi Research Institute, Inc. (Tokyo, Japan).

**Institutional Review Board Statement:** Institutional Review Board Statement: The experimental protocol of this study was approved by the local institutional review board of the National Institute of Advanced Industrial Science and Technology (HF2017-770B).

**Informed Consent Statement:** Informed consent was obtained from all subjects involved in the study.

**Data Availability Statement:** Data sharing not applicable.

**Conflicts of Interest:** The funders had no role in the design of the study; in the collection, analyses, or interpretation of data; in the writing of the manuscript, or in the decision to publish the results.

## Appendix A. Questionnaire

Since all participants were Japanese, all questions were asked in Japanese. Participants answered the following questions using a 5-point Likert scale.

*Appendix A.1. Questions for Each Condition*

Q1  Was the HMD fixed to your head?
Q2  Did you see the graphics clearly?
Q3  Did you feel a physical load from the graphics?
Q4  Did you feel a physical load from the weight?

Q5　Did you feel a physical load from the balance?

*Appendix A.2. Question after All Games Are Finished*

Q1　Do your eyes feel uncomfortable?
Q2　Did your eyes feel more uncomfortable after taking off the HMD?
Q3　Does your head feel uncomfortable?
Q4　Did your head feel more uncomfortable after taking off the HMD?
Q5　Does your neck feel uncomfortable?
Q6　Did your neck feel more uncomfortable after taking off the HMD?

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
