# Peer review of "Effects of the Weight and Balance of Head-Mounted Displays on Physical Load"

_applsci, doi:10.3390/app11156802_

Round 1

Reviewer 1 Report

In this manuscript, researchers have described the interesting studies on the effects of the weight and balance of HMD on physical load. HMD is an emerging technology for AR/VR and mixed reality. The technical writing has been prepared carefully. The literature search is also supporting the main content well. "accepted" is suggested.

Author Response

Thank you for inviting us to submit a revised draft of our manuscript. We also appreciate the time and effort you and each of the reviewers. Thus, it is with great pleasure that we resubmit our article for further consideration. We have incorporated some changes that reflect the suggestions that other reviewers provided. The changes are shown in red. Again, thank you for giving us the opportunity to strengthen our manuscript with your valuable comments and queries. We have worked hard to incorporate your feedback and hope that these revisions persuade you to accept our submission.

Reviewer 2 Report

Overall, the paper is well organized and written. However, I suggest facing the following issues to improve the quality.

Perhaps, it is more appropriate to talk about Center of Gravity as the (dis)comfort depends on the position of the gravity force acting on the camera.

I suggest to better explain the original aspects the work investigates, with respect to the germane literature. In the Introduction, the authors made a great job in summarizing relevant contributions already available. However, most of the recalled contributions already study relationships between loads and (dis)comfort. Therefore, which is the original element of knowledge about the problem that the authors add to the literature with this work? What does it mean that the study reproduces natural situations?

To easy the readability, I suggest adding a brief overview of the content organization in the end of the introduction section.

In reference to the experiment, the parameters measured are clear however it is not clear their purpose. The authors should explain the experimental objectives and subsequently claim which objectives have been assessed through which parameters. Example: the authors “measured the pupillary distance and grip strength of the participants” to the purpose of what? This is also for position and posture of the targets in the shooting game, the corresponding head movements, the time, and the parameters assessed by the questionnaire. About the latter, I suggest adding the questions in a appendix (maybe), otherwise the experiment cannot be replicated by readers that are interested in using the same approach.

For what concern the participants, did the experiment respect the main ethical issues? How did the authors treat data? Was the experiment anonymous? How did the authors enroll the participants? Was the participation voluntary? Did the participants receive a payment or other kinds of benefits? I suggest not neglecting these aspects in the description of the experiment.

The authors used ANOVA as statistical test to assess significant differences among groups. I suggest being more precise about the outcomes of these assessments. The Fisher and Snedecor parameters are essential to demonstrate the significance of differences, the authors should add them. Furthermore, I would remember that the use of ANOVA requires the verification of some hypothesis like, for instance, the normal distribution of the data. Did the authors do this verification? In the figures from 5 to 11 there are some asterisks, which is their meaning? In the figures’ caption, it is not explained. Eventually, the way to build the map presented in Figure 12 is not so clearly explained, I suggest adding more details.

The discussion is adequate and, potentially, well supported by the results (although in the current form important statistical parameters lack to demonstrate the significance of differences). 

Author Response

Thank you for inviting us to submit a revised draft of our manuscript. We also appreciate the time and effort you and each of the reviewers have dedicated to providing insightful feedback on ways to strengthen our paper. Thus, it is with great pleasure that we resubmit our article for further consideration.

We have incorporated changes that reflect the detailed suggestions you have graciously provided. We also hope that our edits and the responses we provide below satisfactorily address all the issues and concerns you and the reviewers have noted.

Point 1: Perhaps, it is more appropriate to talk about Center of Gravity as the (dis)comfort depends on the position of the gravity force acting on the camera.

Response 1: In this study, “Center of Mass” and “Center of Gravity” mean almost the same thing. As you pointed out, the "Center of Gravity" may be more appropriate to talk about the subjective (dis)comfort. However, we would like to use "Center of Mass (COM)" because we think it is easier to understand if we use the same word as in the previous study by Chihara and Seo. Accordingly, we have added the following text to the Introduction (p.2, lines 70-73): In such gameplay, physical workload may depend on the position of the gravity force acting on the camera, rather than on its center of mass. In this study, we use the Center of Mass, which was used in the study by Chihara and Seo. These meanings are almost the same.

Point 2: I suggest to better explain the original aspects the work investigates, with respect to the germane literature. In the Introduction, the authors made a great job in summarizing relevant contributions already available. However, most of the recalled contributions already study relationships between loads and (dis)comfort. Therefore, which is the original element of knowledge about the problem that the authors add to the literature with this work? What does it mean that the study reproduces natural situations?

Response 2: Our original element is performing experiment in natural user situation and investigated the relationship among physical workload, weight and balance. Most important point of "Natural situation" is that the participants get the spontaneous postures during game play. We have added the following text to the Introduction (p.2, lines 52-60): Some studies focused on the comfort of HMDs, but few comprehensively investigate weight and balance. The consumer HMDs are already made lighter in weight, but different weight balances should be considered. Chihara and Seo investigated it. However, the participants held the instructed postures for five seconds with professional heavy HMD. If we assume a natural HMD usage situation, the participants should get the spontaneous postures for at least a few minutes. Therefore, we decided to vary the weight and balance of the lightweight HMD to investigate the physical workload by the participant's spontaneous posture change during game play.

Point 3: To easy the readability, I suggest adding a brief overview of the content organization in the end of the introduction section.

Response 3: We agree that this point requires clarification, and have added the following text to the Introduction (p.2, lines 65-68): In this game, participants shot targets that appeared at random positions and postures within their field of view. The aim of the gun moves in response to changes of the participant's posture. It took around 2 minutes.

Point 4: In reference to the experiment, the parameters measured are clear however it is not clear their purpose. The authors should explain the experimental objectives and subsequently claim which objectives have been assessed through which parameters. Example: the authors “measured the pupillary distance and grip strength of the participants” to the purpose of what? This is also for position and posture of the targets in the shooting game, the corresponding head movements, the time, and the parameters assessed by the questionnaire. About the latter, I suggest adding the questions in a appendix (maybe), otherwise the experiment cannot be replicated by readers that are interested in using the same approach.

Response 4: The purpose of the measured parameters was not fully explained. We have added the following text to the Measurements and Results:

(p.4, lines 110-113)

Pupillary distance and visual acuity were measured to verify low visual acuity or pupillary distance adjustment would interfere with game play. Grip strength was measured as a surrogate for the participant's total body muscle mass.

(p.4, lines 118-119)

These were measured to verify that the game was not progressing due to abnormal behavior.

(p.4, lines 125-127)

The first two questions were asked to verify the presentation of the game images was appropriate. The latter three questions were asked to assess subjective physical workload.

(p.4, lines 132-134)

These questions were asked to verify that the physical workload caused by the weight used in this study would not cause sequelae.

(p.5, lines 136-138)

Based on the measurement results of visual acuity and pupillary distance, all subjects could see our game image normally. In addition, we verified the data collected by the HMD, and there was no abnormal behavior in the game.

In addition, I added a questionnaire to the appendix A.

Point 5: For what concern the participants, did the experiment respect the main ethical issues? How did the authors treat data? Was the experiment anonymous? How did the authors enroll the participants? Was the participation voluntary? Did the participants receive a payment or other kinds of benefits? I suggest not neglecting these aspects in the description of the experiment.

Response 5: We agree that this point is necessary, and have added the following text to the Participants (p.3, lines 102-105): Most of participants was visitors to the exhibition and voluntarily participated. The shortage of participants was recruited through a recruiting company for a fee. The experiment was conducted with the approval of the Ethical Review Committee, and the data were anonymized for statistical analysis.

Point 6: The authors used ANOVA as statistical test to assess significant differences among groups. I suggest being more precise about the outcomes of these assessments. The Fisher and Snedecor parameters are essential to demonstrate the significance of differences, the authors should add them.

Response 6: As you pointed out, we need to add description of statistical parameters. We added detailed statistical results such as F-value to the Results. (p.5-6).

Point 7: Furthermore, I would remember that the use of ANOVA requires the verification of some hypothesis like, for instance, the normal distribution of the data. Did the authors do this verification?

Response 7: We agree to add about the verification of some hypothesis, and have added following text to the Results (p.5, lines 147-148): In the following sections, we performed statistical analysis. ANOVA was performed with a data which has a normal distribution and equal variances.

Point 8: In the figures from 5 to 11 there are some asterisks, which is their meaning? In the figures’ caption, it is not explained.

Response 8: We have added the meaning of the asterisk to the figure caption (Figure. 2-9).

Point 9: Eventually, the way to build the map presented in Figure 12 is not so clearly explained, I suggest adding more details.

Response 9: The explanation of Figure 12 was insufficient. We have added following text to the Results (p.6, lines 204-209): The value on the vertical axis is larger as the weight increases, and on the horizontal axis is larger as the center of mass moves forward from the tragion notch. Several points determined from the experimental conditions are plotted, and the larger the fatigue score, the redder the color. These points were then linearly interpolated to form a contour plot. The blank areas were not included in the experimental conditions.

Again, thank you for giving us the opportunity to strengthen our manuscript with your valuable comments and queries. We have worked hard to incorporate your feedback and hope that these revisions persuade you to accept our submission.

Reviewer 3 Report

This paper presents interesting aspects of interactions between weight and balance of HMDs and users fatique.

The paper's structure is proper and fits journal's requirements. Introduction is sufficient and lets readers to be familiarize with the topic. Material and methods section is constructed with all scienttific standards as weel as statistical methods.

Results are shown cleary with charts, wide described and discuss with others authors experience.

Author Response

(The authors gave the same response as above.)

Reviewer 4 Report

In this way, I think that other tests should be carried out to evaluate the anatomical and physiological effect produced by HMD (effect on the joint, degree of inflammation ...). For example, some kind of imaging test, determination of inflammatory biomarkers, etc. Only a questionnaire is something too subjective. Furthermore, the elderly population involves a bias in the study as a consequence of their less familiarity with video games. This problem should be solved with the objective that the time factor is similar in all groups.

Author Response

Thank you for inviting us to submit a revised draft of our manuscript. We also appreciate the time and effort you and each of the reviewers have dedicated to providing insightful feedback on ways to strengthen our paper. Thus, it is with great pleasure that we resubmit our article for further consideration.

We have incorporated changes that reflect the detailed suggestions you have graciously provided. We also hope that our edits and the responses we provide below satisfactorily address all the issues and concerns you and the reviewers have noted.

Point 1: In this way, I think that other tests should be carried out to evaluate the anatomical and physiological effect produced by HMD (effect on the joint, degree of inflammation ...). For example, some kind of imaging test, determination of inflammatory biomarkers, etc. Only a questionnaire is something too subjective.

Response 1: As you pointed out, it is the limitation of this paper. More objective evaluations is definitely our future work. We have added the following text to the Discussion (p.11, lines 291-294): Our results are based on subjective responses. More objective evaluations are needed to verify the anatomical and physiological effects. For example, future work is to estimate the load torque applied to the neck joint from the posture and weights of human and HMD, and compare it with the present results.

Point 2: Furthermore, the elderly population involves a bias in the study as a consequence of their less familiarity with video games. This problem should be solved with the objective that the time factor is similar in all groups.

Response 2: The familiarity factor may introduce a bias that does not reflect the full extent of the effects of HMD weight and balance. However, in this experiment, we designed this experiment to evaluate with representative users of each age group. We considered that the bias due to lack of familiarity with games and an age bias may be inseparable.

Accordingly, we have added the following text to the Discussion (p.9, lines 233-237): We designed this experiment to clarify the differences in the appropriate weight and balance of HMDs depend for each age. There is a bias as a consequence of their less familiarity with video games. However, if the elderly users are generally not familiar with HMD-based games, such participants should be targeted. Therefore, we considered that the bias due to lack of familiarity with games and an age bias may be inseparable.

Again, thank you for giving us the opportunity to strengthen our manuscript with your valuable comments and queries. We have worked hard to incorporate your feedback and hope that these revisions persuade you to accept our submission.

Round 2

Reviewer 2 Report

The authors faced all the issues and, according to my review, the paper is now worthy of publication.

Reviewer 4 Report

As preliminary results, it may be acceptable but more objective evaluations are needed to verify the anatomical and physiological effects in future studies.